# Integrated Analysis of Long Non-Coding RNA and mRNA Expression Profiles in Testes of Calves and Sexually Mature Wandong Bulls (*Bos taurus*)

**DOI:** 10.3390/ani11072006

**Published:** 2021-07-05

**Authors:** Hongyu Liu, Ibrar Muhammad Khan, Huiqun Yin, Xinqi Zhou, Muhammad Rizwan, Jingyi Zhuang, Yunhai Zhang

**Affiliations:** 1College of Animal Science and Technology, Anhui Agricultural University, Hefei 230036, China; liuhongyu@ahau.edu.cn (H.L.); ibrar.pesh@gmail.com (I.M.K.); xinqizhou2021@163.com (X.Z.); rizwiktk007@hotmail.com (M.R.); Zhuang_JY320@163.com (J.Z.); 2Anhui Provincial Laboratory of Local Livestock and Poultry Genetic Resource Conservation and Breeding, Anhui Agricultural University, Hefei 230036, China; 3Reproductive Medicine Center, The 901st Hospital, Hefei 230031, China; milklecherry@163.com

**Keywords:** testicular growth, spermatogenesis, mRNAs and lncRNAs axes, total RNA-sequencing

## Abstract

**Simple Summary:**

The Chinese native cattle breeds are economically important, and these breeds are used extensively for beef production. However, their reproductive efficiency and the local elite genetic breeding populations, which is an important consideration in livestock production systems, have not been fully established yet. The literature recommends the application of molecular strategies that improve the production of good-quality sperm and increase the availability of elite genetics for improving local breed production. Therefore, this study comprehensively analyzed the mRNAs and long non-coding RNAs expression profiling, based on total RNA-Seq, of the testis samples of bulls of two distinct age groups. The findings of this study have strong implications, as they provide a suitable framework for understanding the mechanism of mRNAs and long non-coding RNAs in the development of testes and spermatogenesis.

**Abstract:**

The mRNAs and long non-coding RNAs axes are playing a vital role in the regulating of post-transcriptional gene expression. Thereby, elucidating the expression pattern of mRNAs and long non-coding RNAs underlying testis development is crucial. In this study, mRNA and long non-coding RNAs expression profiles were investigated in 3-month-old calves and 3-year-old mature bulls’ testes by total RNA sequencing. Additionally, during the gene level analysis, 21,250 mRNAs and 20,533 long non-coding RNAs were identified. As a result, 7908 long non-coding RNAs (*p*-adjust *<* 0.05) and 5122 mRNAs (*p*-adjust *<* 0.05) were significantly differentially expressed between the distinct age groups. In addition, gene ontology and biological pathway analyses revealed that the predicted target genes are enriched in the lysine degradation, cell cycle, propanoate metabolism, adherens junction and cell adhesion molecules pathways. Correspondingly, the RT-qPCR validation results showed a strong consistency with the sequencing data. The source genes for the mRNAs (*CCDC83, DMRTC2, HSPA2, IQCG, PACRG, SPO11, EHHADH, SPP1, NSD2* and *ACTN4*) and the long non-coding RNAs (*COX7A2, COX6B2, TRIM37, PRM2, INHBA, ERBB4, SDHA, ATP6VOA2, FGF9* and *TCF21*) were found to be actively associated with bull sexual maturity and spermatogenesis. This study provided a comprehensive catalog of long non-coding RNAs in the bovine testes and also offered useful resources for understanding the differences in sexual development caused by the changes in the mRNA and long non-coding RNA interaction expressions between the immature and mature stages.

## 1. Introduction

Improvement in the reproductive efficiency of local cattle bull via genetic selection is crucial, as individual bulls can breed multiple numbers of cows [1]. Advances in livestock genomics, such as transcriptomics, proteomics and single-nucleotide polymorphism genotypes studies, are predicted the direct increasing rate of genetic gain in a generation [2,3]. Thus, it demands an extensive use of molecular tools to improve the reproductive traits of Chinese native cattle.

Mammalian spermatogenesis takes place by the subsequent division of the cell: self-restoration of gonocytes, mitotic proliferation of spermatogonia, meiotic division of primary spermatocytes and post-meiotic differentiation of secondary spermatocytes. Finally, the haploid spermatids transform into another process called spermiogenesis [4]. The process of spermatogenesis is more curious and involves several transcriptional factors that regulate by the testis-specific genes at the transcriptional and post transcriptional levels [5,6]. The interactive network of lncRNAs and mRNAs is thought to be involved in mammalian reproduction system regulation and sperm quality parameters [7].

The lncRNA profiles are comprehensively analyzed and identified at the 3-day-old neonatal and 13-month-old mature stages of Angus cattle [8]. The detailed catalog of known and novel lncRNAs was also identified at four different developmental stages, including 60, 90, 120 and 150 days in boar testes [9]. Non-coding RNAs (ncRNAs) with biological functions were identified nearly 60 years ago, and other functional ncRNAs were considered to occur as early as the 1980s [10]. Later, ncRNAs such as Xist were discovered to control the chromosome structure. Since then, the number of novel and potentially usable ncRNAs has grown dramatically [11,12]. However, new literature suggests that ncRNAs such as microRNAs [12], PIWI-interacting RNAs, small interfering RNAs and long non-coding RNAs (lncRNAs) play a wide range of functions [13,14,15,16]. These massive quantities of transcript species are referred to as ncRNAs.

In several animal species, the identification and functional validation of lncRNAs have been performed and characterized in spermatogenesis and testicular development [17,18]. Many lncRNAs have been found in mouse (e.g., 8265 lncRNAs and 18,563 mRNAs), Drosophila (128 testis-specific lncRNAs), pig (15,528 lncRNAs), chicken (2597 lncRNAs and 17,690 mRNAs) and in sheep (6460 lncRNAs and 42,300 mRNAs) at various stages of testis development and spermatogenesis [19,20,21,22,23].

Recently, lncRNAs have been recognized in various developmental stages of testes and spermatogenesis in rat, mouse and human models. The lncRNA link genes Membrane protein related to HSP30 (*MRH1*), testis-specific X-linked gene (*TSX*), DNA methylation regions (*DMR*) and HONGRES2 are predicted to have critical roles in testicular growth and spermatogenesis [24,25]. The *DMRT1* protein has been shown with a role in spermatogonial growth and avoiding premature meiosis in spermatogonia, implying that the lncRNA Dmr could be involved in the transition between mitosis and meiosis during spermatozoa development [26]. The testis-specific X-linked lncRNAs were found to be expressed exclusively in pachytene spermatocytes but not in the spermatogonia or round spermatids, implying a regulatory role in germline meiotic division [23].

We investigated the supportive network of lncRNA-mRNAs and their expression profiles in immature and sexually mature bull testes. Our research identified the lncRNAs-mRNA integrative network during the sexual maturation of bulls (*Bos taurus*) and provided a useful landscape resource for reviewing the useful roles of ncRNA in testis development and spermatogenesis. We performed transcriptome sequencing and examined the lncRNA-mRNA expression design in the testis. The obtained results will provide potential molecular fingerprints for assessing the developmental status of bull testes and, also, facilitate the breeding soundness examination in local elite bulls for the AI program.

## 2. Materials and Methods

### 2.1. Experimental Animals and Sample Collection

Wandong cattle are an autochthonous Chinese breed used extensively for beef production. They tolerate extreme weather conditions and raw feed and are resistant to tick-borne diseases. Two different age group (*n* = 3) of physically healthy and sexually mature Wandong bulls (3 ± 0.014 years) and immature calves (3 ± 0.24 months) were selected for the current research [27]. The sexually mature group was subjected for seminal quality traits, and the semen volume, concentration, grass motility, progressive motility and spermatozoa defects were measured by specialized technicians (Appendix A). Andrology testing was also carried out, where the selected bulls were kept in another pen to mount other bulls and become aroused. The bulls and calves were slaughtered for testicle samples collection at Fengyang County, Anhui, China. A total of six samples of testes from immature calves and mature Wandong bulls (*Bos taurus*) were collected (Appendix A). All pairs of testes were processed by incising the scrotum medially and uncovering the right and left testicles within the tunica vaginalis. The samples were immediately put in a cryogenic vial and frozen in liquid nitrogen (−195.8 °C) until use. The collection of experimental animals was carried out under strict ethical procedures and approved (SYXK 2016-007) by the Animal Care Unit and College of Animal Science and Technology, Anhui Agricultural University. All possible efforts were made to minimize any pain experienced by the experimental animals.

### 2.2. Histological Study of Testicular Tissues

The bulls and calves testes tissue samples were washed with 0.9% and preserved with Bouin’s solution containing 75-mL saturated bitter acid solution, 25-mL formaldehyde and 5-mL glacial acetic acid for 48 h at room temperature [28]. Next, the tissues were embedded in paraffin and dissected at parts of 6-μm-thick sections and stained with hematoxylin–eosin (HE). The histomorphology of the testicular tissues was analyzed using 100×, 200× and 400× magnifying light microscopies by Nikon Eclipse C1; Nikon, Tokyo, Japan.

### 2.3. RNA Isolation and Integrity Assessment

The testicular samples were sent to a commercial sequencing facility (Beijing Novogene Corporation, Beijing, China) for Seq-analysis. Total RNA was isolated from the testis samples using Trizol reagent (Takara, Beijing, China) in sterile conditions and treated with DNase I enzyme to remove endogenous DNA contamination according to the manufacturer’s protocols. RNA quality and contamination were assessed by 1% agarose gel electrophoresis. The purity was checked using a NanoPhotometer^®^ spectrophotometer (IMPLEN, Westlake Village, CA, USA). The total RNA concentration was assessed using the Qubit^®^ RNA Assay Kit in Qubit^®^ 2.0 Fluorometer (Life Technologies, Carlsbad, CA, USA). RNA integrity was measured using the RNA Nano 6000 Assay Kit of the Bioanalyzer 2100 system (Agilent Technologies, Santa Clara, CA, USA).

### 2.4. cDNA Library Construction for lncRNA-mRNAs Sequencing Analysis

A total of 3μg of RNA from each testis sample of 3-month-old calves and 3-year-old bulls were used in construction of the cDNA library. In the present study, six (calves: *n* = 3 and bulls: *n* = 3) seq-libraries were generated by using the NEB-Next^®^ Ultra-TM RNA Library Prep Kit for Illumina^®^ (NEB, San Diego, CA, USA). The Novogene Bioinformatics Institute constructed a chain-specific library by removing the ribosomal RNA [28]. First of all, the ribosomal RNA was removed from the total RNA, and then, the RNA was divided into many short slices of 250–300 bp. The first cDNA as a template strand was manufactured from fragmented RNA and the random use of oligonucleotides as primers. In the next step, the RNA strand was degraded with RNase H, and then, the second cDNA strand was manufactured along with dNTPs (dATP, dGTP, dUTP and dCTP) as a material underneath the DNA polymerase I system. After the purification, double-stranded cDNA was fixed while followed by a tail step and connected to the sequencing connector. A cDNA around 200 bp was separated with AM-Pure XP beads. The USER enzyme degraded the second strand of the cDNA, which contained U, and PCR amplification was performed to obtain the library. Qubit 2.0 (https://www.thermofisher.com, accessed on 11 April 2019) was used to quantify the library, which was then diluted to 1.5 ng/uL, whereas the total insert size of the library was detected by Agilent 2100. When the library qualified the inspection process, the Illumina PE150 (Hi-Seq 4000 System) (https://www.ietltd.com//biotech, accessed on 11 April 2019) sequencing machine was run to identify the data output demand, and 150-base pair (bp) paired-end reads were generated. The cDNA library construction and quality inspection are shown in Appendix A.

### 2.5. Transcriptome Assembly

After isolation of the reads with adaptors and a poly-N greater than 10% and poor-quality reads from raw data using in-house perl scripts produced by the Bioinformatics Institute (Beijing, China), clean transcriptomic data were obtained. The ribosomal RNAs free reads of each calf and bull sample were charted to reference genome *Bos taurus*-UMD 3.1.1 by TopHat2 (v. 2.0.3.12), as described in reference [29,30]. Bowtie v2.0.6 was used to build an index of the bovine reference genome [31], and the paired-end clean reads were aligned using TopHat [32] v2.0.9 (https://ccb.jhu.edu/software/tophat/index.shtml, accessed on 11 November 2019). The mapped reads of each sample were assembled by both Scripture (beta2) [33] and Cufflinks v2.1.1 [34].

### 2.6. LncRNAs and mRNAs Identification and Coding Potential

LncRNA has a variety of classification methods, and Novogene references HGNC to classify the newly screened lncRNAs into the following four types, according to their positional relationship with known mRNAs. In order to cut down the false discovery rate (FDR), we adopted these subsequent steps to classify the diverse classes of lncRNAs, including lincRNA, intronic lncRNA, antisense lncRNA and sense-overlapping [35,36]. All the assembled transcripts of the six sequencing libraries were associated when using cuffcompare software [37], and those transcripts that were assembled by Scripture (beta2) [33] were discarded. The transcripts possessing only single exon and less than 200 bp were also discarded, and the reads coverage of every transcript was designed using Cufflinks v2.1.1 (http://cole-trapnell-lab.github.io/cufflinks/releases/v2.1.1/, accessed on 12 February 2019) [33]. All those transcripts that had less than three-reads coverage and an expression value (FPKM) less than 2 (log2-fold change) were removed, and the remaining known transcripts were blasted with bovine known lncRNAs in ALDB [38] by cuffcompare software. Coding potential of lncRNAs was determined by coding non coding Index (CNCI) software (https://github.com/www-bioinfo-org/, accessed on 5 June 2020) [39], Pfam-scansoftware (https://www.ebi.ac.ukpfamscan/, accessed on 5 June 2020) [40,41] and coded potential calculator (CPC) software (http://cpc2.gao-lab.org, accessed on 5 June 2020) [42]. As a result of this software, all transcripts with coding potential were categorized as putative lncRNAs, whereas those lacking coding potentials were classified as non-coding RNAs. After filtering out, the database of known transcripts was categorized into a novel class of lncRNAs and mRNAs, and the different steps are shown in Appendix A.

### 2.7. Expression Pattern of Differentially Expressed lncRNAs and mRNAs

The lncRNA and mRNA gene expression levels were assessed using the FPKMs (fragments per kilobase of transcript sequence per millions of base pairs sequenced) value. A significant analysis of the expression differences (ED) at the gene or transcript level was done, and the functional genes or transcripts relevant to the testis developmental process were found. The Novogene Bioinformatics Institute (Beijing, China) used Cuffdiff (v2.1.1) (https://bioinformaticshome.com/tools/rna-seq/descriptions/Cufflinks.html, accessed on 11 November 2019) software and projected the FPKM levels of the lncRNAs and mRNAs [34]. The DE genes were considered statistically significant by a log2-fold change greater or equal to 2 (*p*-value < 0.05) or *p*-adjust < 0.05 using Ballgown [43].

### 2.8. GO and KEGG Pathways Enrichments

The GO enrichment analysis provided significant source genes that involved different biological functions, and lncRNA DE genes were employed by the GO-seq R package [44]. All the source genes in the pathways and the background genes were charted by GO terms into the database (http://www.geneontology.org/, accessed on 11 November 2019). GO terms with corrected *p* < 0.05 were considered significantly enriched by DE genes as per the definition of the hypergeometric test. Following that, the term enrichment significance analysis was corrected using the false discovery rate (FDR), and the corrected *p*-value (Q-value) was acquired [45]. The KEEG pathways-based study added an explanation of the source genes and their biological functions (http://www.genome.jp, accessed on 11 November 2019). KOBAS 2.0 (http://kobas.cbi.pku.edu.cn, accessed on 11 November 2019) software was suggested by reference [46] to test the enriched DE source genes in the KEGG pathways. The enrichment analysis significantly identified the signaling and metabolic pathways where the lncRNAs source genes and total background genes were charted.

### 2.9. Cis and Trans-Regulated Genes Prediction

A classic instance of *trans*-acting is transcription factors and lncRNAs [47], and DE lncRNAs were chosen for forecasting target genes. Increasing proof has shown that lncRNAs can impact the expression of neighboring coding genes through a *cis*-regulating role. We sought to discover the coding genes located both 10 and 100 Kb upstream and downstream of the lncRNAs and figured out their *cis*-functions [48]. The *trans*-role in lncRNAs is to communicate by the level of expression with target genes. The expressed correlation between lncRNAs and coding genes with custom scripts and target genes identified by a Pearson correlation coefficient >0.95 was calculated [49].

### 2.10. Validation of DE lncRNAs and mRNAs Gene via RT-qPCR

Whole RNA was extracted from the immature and mature groups of testes tissues using a Trizol reagent (Life Technologies, USA), and its concentration was measured using a Nanodrop spectrophotometer. The good-quality RNAs were then converted into cDNAs using a QuantiTect Reverse Transcription Kit (Qiagen, Hilden, Germany) in accordance with the manufacturer’s protocols. The software programs Primer 3 web version 4.0.0 and Basic Local Alignment Search Tool (BLAST; https://blast.ncbi.nlm.nih, accessed on 11 June 2020) were used to design gene-specific primers. The primers used in this study are presented in Appendix A. The PCR primers were generated using FastStart SYBR Green Master Mix (Roche, Germany) and the StepOne Plus Real-Time PCR System (Applied Biosystems, Waltham, MA, USA). Each PCR reaction comprised of 7.5-μL 2×SYBR Green PCR Master Mix, 1.5-μL cDNA, 0.5 μL each of the reverse and forward primers, 4.7-μL nuclease-free water and 10.3 μL of ROX dye. The following q-PCR thermal cycles were carried out: (I) pre-denaturation at 95 °C for 10 min, (II) followed by 45 denaturation amplification cycles at 95 °C for 15 s, (III) annealing at 60 °C for 10 s and (IV) an extension cycle at 72 °C for 20 s. The quantification cycle (Cq) of each target gene was normalized to that of the reference *GAPDH* gene. The q-PCR experiment was performed in triplicate to minimize the risk of error in the experiments. The Cq values were obtained and transferred to a Microsoft Excel sheet for a further relative quantification analysis using the 2^−ΔΔ*C*t^ method [50].

### 2.11. Statistical Analysis

Data on the lncRNA and mRNA transcripts of interest from immature and mature testicular tissues were analyzed using the Student’s *t*-test (SPSS 17.0) and presented as the mean ± SEM. Before conducting the *t*-test, the data distribution and variances between the two groups were evaluated and found to be normal and homogenous, respectively. The mean values were believed to be significantly different with * *p* < 0.05 and ** *p* < 0.01.

## 3. Results

### 3.1. The Morphological Study of Testicular Tissues

The histomorphology of the bull (*Bos taurus*) testes showed a noteworthy difference during the immature and mature stages. Under a 100× microscopic study, the diameter of the seminiferous tubules was much smaller in the immature as compared to the mature testes. At the same time, similar circumstances were found in the interstitial connective tissue of the immature and mature testes. Under a 400× microscopic study, we got further details and found more developmental stages of biological spermatozoa in the mature testes than the immature. The Sertoli cells were significantly increased in size and found very near to the basement line of the seminiferous tubules in the mature testes, as shown in Appendix A.

### 3.2. Outline of lncRNAs and mRNAs Sequencing in Calf and Bull Testes

The raw reads were filtered and classified into different segments; consequently, 98.95% (70,333,218 clean reads) of the sequencing data was screened, containing N 0.05% (38,769) reads, low-quality 0.18% (126,760) reads and adopter-related 0.82% (584,096) reads for the bulls, and similar classifications were done for the calve, including clean reads (70,218,119, 98.48%), containing N (37,396, 0.05%), low-quality (141,228, 0.20%) and adopter-related (905,921, 1.27%) (Appendix A). The raw reads for the total samples were found between 118,785,372 and 142,165,686, while the clean reads per sample were 116,004,150 and 140,666,436. Thus, the raw reads and clean reads yielded together 90 GB data, and the GC contents were 47.09–53.57% (Figure 1a). The sequencing process itself has the possibility of machine errors. The distribution check of the sequencing error rate can reflect the quality of the sequencing data, and the sequencing error rate distribution test is used to detect the error % in the sequencing of samples (Figure 1b). All the % Q20 values of the subjected reads in six samples were surpassed by 97%, and all the details are shown in Appendix A. More than 95.5% of the clean reads were aligned to *B taurus* UMD3.1 and are shown in (Figure 1c).

### 3.3. Identification and Characterizations of lncRNAs and mRNAs

We classified and identified lncRNAs based on the position and direction of their genomic regions relative to mRNAs. A total lnc-mRNA magnitude of 44,279 was found in the calf and bull testes, where, at the gene level analysis, 21,250 mRNAs and 20,533 lncRNAs are identified (Figure 2a). We established an accurate putative lncRNAs and mRNAs network from the assembled transcripts with a stringent pipeline. Based on the structural characteristics and genomic position, the lncRNAs were classified into four subclasses, though denoted as lincRNA (75.4%), antisense (17.6%), sense-overlapping (7.0%) and sense-intronic (0.0%) (Figure 2b). The lncRNAs were overlapped in same or in the opposite direction with the exonic or intronic parts of the coding gene; thus, they became exonic-sense, exonic-antisense, intronic-sense or intronic-antisense, respectively. The genomic locus that was more than 1 kb from any mRNA-encoding genomic region was considered as a long intergenic ncRNA (lincRNA). According to this classification scheme, we found that most testis-specific lncRNAs (75.4%) were lincRNA. We examined the chromosomal localization of the testicular lncRNAs and mRNAs and identified the loci that expressed these transcripts. Our research confirmed that the transcripts were uniformly spread throughout the chromosomes of the bulls and calves (Figure 2c). The coding potential of the lncRNA and mRNA were analyzed by using CPC2, CNCI and Pfam software, and a total of 35,409 coding transcripts were identified (Figure 2d).

### 3.4. Characteristics Comparison between lncRNAs and mRNAs

We verified the general characteristics of the novel lncRNAs and mRNAs by comparing the transcript length, exon number and ORF length, which also differentiated between the lncRNAs and mRNAs. The number of exons of mRNAs about (30) was higher than of the lncRNAs (10). The majority of lncRNAs have three or less than three exons, whereas mRNAs have five or more exons (Figure 3a). In the length comparison, we observed that the lncRNAs (an average of 2000 bp) were significantly shorter than the mRNAs (an average of 3000 bp), as shown in (Figure 3b). The open reading frame (ORF) is the normal nucleotide sequence of a structural gene. From the initiation codon to the termination codon, the open reading frame encodes a complete polypeptide chain, without any interruption in the translation. When we compared the lncRNA with mRNA ORFs, the sequences of annotated mRNAs and annotated lncRNAs were extracted by annotations of the known gene structure. The obtained ORF sequence was converted into a protein sequence, and the length distribution was obtained (Figure 3c).

### 3.5. Differentially Expressed Gene-Level lncRNAs and mRNAs in Testicular Tissues

After the quantification process, the expression pattern of the lncRNAs and mRNAs were identified through the cuff-diff and Ballgown tools. The average transcripts expression level was higher in the bulls compared to the calves group (Figure 4a). Thus, we recorded the differentially expressed (DE) lncRNAs and mRNAs between immature and mature testes by using Ballgown. The significant levels of lncRNAs and mRNAs among the testes samples were calculated, and the parameter of significance was taken into account, whereas the log2-fold change was considered higher than two or equal to and *p*-adjusted *<* 0.05. As a result, 7908 lncRNAs and 5122 mRNAs were found to be differentially expressed (DE) in immature and mature bovine testes. Further, we identified 6849 lncRNAs substantially upregulated and 1058 are downregulated in the testes (*p*-value *<* 0.05). In the adult testes group, *p*-adjust *<* 0.05 revealed that 2857 mRNAs were upregulated and 2264 were downregulated. These up and down highlights are displayed in volcanic plots (Figure 4b,c), respectively. We also analyzed the DE lncRNAs and mRNAs by using the hierarchical clustering method, and the clustering was another way to display the DE genes, which bring together genes with similar expression patterns that may have common functions or participate in common metabolic and signaling pathways. The genes cluster on the left side was made due to similar expressions (fold change >2, *p* < 0.05), and the calves and bulls were represented by columns, while the expression from blue to red represented them gradually upregulated (Figure 4d,e).

### 3.6. Functional Annotation and Enrich KEGG Pathways Analysis of DE mRNAs

The transcriptome level examination showed 3857 DE mRNAs (2352 upregulated and 1504 downregulated) were marked, and these mRNAs were preferred for the enrichment analysis. The DE mRNAs were substantially enriched in GO terms, including the reproduction, male gamete generation, spermatogenesis, multicellular organism reproduction, single-organism reproductive process, spermatids differentiation, spermatids development and sexual reproduction, which were the top eight GO annotations terms (Figure 5a). Detailed descriptions of the GO terms are available in Appendix A. As for as the statistics of the pathway enrichment, the DE mRNAs were enriched in the top twenty KEGG-rich factors pathways, such as adherens junctions, cell adhesion molecules (CAMs) and lysine degradation (Figure 5b). Appendix A represents the top 10 most significant enriched pathways regulated by mRNAs in Wandong testes development. A total of 5122 DE genes (2857 upregulated and 2264 downregulated), which were regulated by mRNAs, were enriched in the GO terms related to reproduction and spermatogenesis. Some genes were (*FSCN3, HOXA11, CCDC155, LOC528479, CCNB1IP1, OCA2, DMRTC2, ADIG* and *PACRG*) highly related to spermatogenesis, while genes like *KIF18A, FNDC3A, KITLG, SOX8, HOXA11, UBB, SFRP1, KLHL10* and *WNT2B* were found actively in male gonad development. The top 20 statistically enriched pathways (*p* < 0.05) were assessed, and the source genes were found in the signaling pathways, such as lysine degradation, cell cycle, propanoate metabolism, adherens junction and cell adhesion molecules (Figure 5c).

### 3.7. LncRNAs Co-Location and Co-Expression Regulated Target Genes

The target gene prediction was performed based on the correlation between the lncRNAs and transcription factors, and the screening condition suggested that the correlation coefficient should be greater than 0.95. For the elucidation of lncRNA functions during the biological process, we predicted the targets of the lncRNAs. For the co-location-related target gene analysis, a *cis*-target gene prediction was performed while screening range was within 100 Kb, and DE *cis*-acting lncRNAs were shown (Table 1). The co-expression network built for bull and calf testes comprised 1168 mRNAs and 410 lncRNAs, which have regulatory potential. Taking 100 Kb as the cutoff, 410 out of 550 lncRNAs were found as the nearest neighbors of 1168 mRNAs. The findings of the GO enrichment study revealed that there were 70 GO descriptions significantly observed (*p*-adj-value *<* 0.05). The top five GO terms were cellular process, cell, cell part, cellular component and regulation of the biological process (Figure 6a). The top 10 significantly enriched GO descriptions of BP, CC and MF, which regulated the target genes of the lncRNAs, are shown in Appendix A. We also studied the co-location target genes in a *trans* way for the lncRNAs. A total of 550 lncRNAs and 11,470 genes were detected using Pearson correlation 0.95 as the cutoff. The GO enrichment analysis for co-located target genes identified 190 significant GO terms (*p* < 0.05), while the top five GO terms were spermatogenesis, sexual reproduction, male gamete generation, gamete generation and reproduction (Figure 6b). The KEGG analysis for co-expression identified 10 pathways (*p* < 0.05), and the top 20 pathways are shown in (Figure 6c), such as the T-cell receptor signaling pathway, Ubiquitin-mediated proteolysis, mRNA surveillance pathway, ECM−receptor interaction, focal adhesion and cell cycle pathway. Appendix A shows the top 10 most significant enriched pathways regulated by lncRNAs in Wandong testes development. Concerning the pathways, the co-located *trans*-target genes for the lncRNAs were supplemented in several KEGG pathways—for example, the ribosome pathway, Lysine degradation, fructose and mannose metabolism, Vitamin B6 metabolism and TGF-beta signaling pathway (Figure 6d).

### 3.8. Validation of Differentially Expressed LncRNAs and mRNAs

We randomly selected 10 DE mRNA-predicted genes (e.g., *CCDC83*, *DMRTC2*, *HSPA2*, *IQCG*, *PACRG*, *SPO11*, *EHHADH*, *SPP1*, *NSD2* and *ACTN4*) and 10 lncRNA-predicted genes (*COX7A2*, *COX6B2*, *TRIM37*, *PRM2*, *INHBA*, *ERBB4*, *SDHA*, *ATP6VOA2*, *FGF9* and *TCF21*) for the immature and mature testes samples, using RT-qPCR for the purpose of validating the genes that identified in RNA-Seq results. The RT-qPCR relative expression results were parallel to the RNA-seq results as shown (Figure 7a–d), whereas the primer details are available in Appendix A. The achieved results demonstrated that the RNA-seq data was valid and dependable, with a specificity in the bull testes developmental stages. To further elucidate the *cis*-acting potential of the lncRNAs and their target genes in the testicular growth of bulls, four candidate lncRNA: TCONS_00119977, TCONS_00034227, TCONS_00059659 and TCONS_00108262, along with cis-acting genes: *EIF4ENIF1*, *ARID4A*, *CDADC1* and *REN* were selected, while the RT-qPCR and the obtained results were in line with the RNA-Seq findings (Figure 7e–h).

## 4. Discussion

In the present research, we studied the DE lncRNAs and mRNAs during the testicular growth of local Wandong cattle and also investigated the potential regulatory roles of lncRNAs and *cis*-acting genes in cattle bull sexual development. The histological transition in testes occurs primarily at three developmental phases: the premature, mature and adulthood periods. Testicle growth is usually followed by spermatogenesis, so we found only spermatogonia, spermatocytes and Sertoli cells in the seminiferous tubules in the 3-month-old group, while spermatids and spermatozoa were also observed in the 3-year-old adult group. The very similar differences were also noticed in 3-day-old neonatal and 13-month-old mature Angus bulls [8]. LncRNAs have received a lot of coverage as a novel regulatory player in cellular growth [51]. LncRNAs are abundant in the testes, and their average number fluctuates in a similar pattern to mRNAs during the developmental changes [39,52,53]. LncRNAs were found, in several experiments, to play a significant role in mammalian testes formation and spermatogenesis. Numerous lncRNAs from testicular tissues or germ cells of mice [54], porcine [21] and humans [35,55] have been identified. Weng et al. [55] isolated 240 proven and 537 putative lncRNAs from the testes of 30- and 180-day-old male Shaziling Chinese pigs.

The accurate putative lncRNAs and mRNAs combined network was assembled and characterized. A total of 21,250 mRNA and 20,533 lncRNA transcripts were identified at two developmental stages of local bulls. Gao et al. [8] identified a total of 23,735 lncRNAs and 22,118 mRNAs in two developmental stages of Angus cattle testes, in which 540 lncRNAs and 3525 mRNAs were differentially expressed (DE) between the neonatal and adult stages. Furthermore, we discovered that the mRNAs and lncRNAs had identical chromosomal distributions at the autosomes and the X-chromosome, suggesting that the lncRNAs interacted closely with the mRNAs. The higher numbers of lncRNAs (6460) and mRNAs (42,300) were also characterized and identified in sheep testes during sexual maturation [23]; similarly, 36,727 lncRNAs were found in the postnatal (30-day and 180-day-old) pig testes [21]. The maximal numbers of both the lncRNAs and mRNAs were observed in the testes compared to the other tissues, suggesting that the testes are characterized by a high level of transcriptomic diversity and complexity [56].

The number of lncRNAs discovered in our study was substantially higher than that observed in bovine skeletal muscle (4155) [57]; from ovaries of Duroc pig (2076) lncRNAs [58], (3481) lncRNAs in milk exosomes [59], in mammary gland (6450) lnc-RNAs [60] and bovine ileum (1568) lncRNAs [61], it is revealing that lncRNAs are testes-specific. We differentiated between lncRNAs and mRNAs via comparing the transcript length, exon number and ORF length. The lncRNA exons were found relatively shorter as compared to mRNAs. We found that this genomic comparison between the lncRNAs and mRNAs was in line with recent studies on other mammals [20,22].

A total of 5122 mRNAs and 7908 known lncRNAs were significantly DE between the immature and mature groups. Further, we identified 6849 lncRNAs significantly upregulated and 1058 downregulated, whereas 2857 mRNAs were upregulated and 2264 were downregulated between the calf and bull groups. A magnitude of 23,735 lncRNAs and 22,118 mRNAs were detected in the testes of Angus cattle at two developmental phases, where 540 lncRNAs and 3525 mRNAs were discovered DE in 3-day-old calves and 13-month-old bulls [8]. Weng et al. [9] identified 15,528 lncRNAs, while these transcripts consisted of 5032 known and 10,496 novel lncRNAs in four different postnatal developments of testes in porcine. Bao et al. [53] investigated the expression profiling of thousands of lncRNAs and mRNAs and hundreds of lincRNAs in male spermatozoa progress and testes development. According to the previous studies, lncRNA expressions in porcine and mouse testes were strongly associated with specific developmental stages [19,21]. Our systematic studies also proposed that the lncRNAs and mRNAs were specifically expressed in the two developmental stages of bovine testes.

To thoroughly analyze the biological functions of lncRNAs and mRNAs in local bull testes, GO and KEGG studies were done for the linked genes of DE lncRNAs and mRNAs. Through these bioinformatic analyses, candidate genes were found, and these genes were linked to male reproduction. The results showed that 247 mRNA transcripts were enriched for the following GO descriptions: biological process, molecular function and cellular component. The source genes linked to mRNAs, such as *CCDC83*, *DMRTC2, HSPA2, IQCG, PACRG, SPO11, EHHADH, SPP1, NSD2* and *ACTN4*, and genes linked to lnc-RNA transcripts, like *COX7A2, COX6B2, TRIM37, PRM2, INHBA, ERBB4, SDHA, ATP6VOA2, FGF9* and *TCF21*, were involved in the GO sub-descriptions like spermatid development, spermatogenesis, spermatid differentiation, reproductive process, male gonad development, cellular process involved in reproduction, sperm motility, inactivation of MAPK activity and steroid hormone-mediated signaling pathway, which was similar to the GO terms analysis in sheep testes [23]. A catalog of candidate genes was discovered in the testes of Angus cattle, and some of these genes were linked to male reproduction. The *FSCN3, TNP2, IQCF1, PRM2* and *MORC1* were related to spermatogenesis. *KLHL10, TCF21* and *SFRP1* may be involved in the development of male gonads and the androgen receptor [8]. Furthermore, the biological pathway reports verified that many DE source genes actively participated in the reproduction-linked signaling pathways, like adherens junctions, cell adhesion molecules (CAMs) and lysine degradation, and these pathways were also associated with male reproduction [62,63].

The cell cycle, lysine degradation, tight junction and adherence junction pathways and the host genes that link to ncRNA transcripts work together in mammalian testes growth, spermatogenesis and primary sexual activity [64]. It is well-known that gene expression profiling regulates testes development and spermatogenesis in mammals; we identified many DE mRNA and lncRNA genes that may take part in bovine reproduction. The candidate genes like the double sex and mab-3-related transcription factors (*DMRT*) and the family C2 (*DMRTC2*) are required for mammalian testicular functions and spermatogenesis, and *DMRTC2* is highly conserved in sheep spermatogenesis and testis growth [65] and also plays a significant role in fox hybrids during spermatogenesis [66]. The known candidate *HSPA2* gene is enriched in testes and is a member of the 70-kDa heat shock protein family that facilitates the folding, delivery and assembly of protein complexes and has been linked to in vitro fertilization (IVF) performances in both humans and rodents [67,68]. These source gene *IQCG* was mainly involved in the tight junction and adherens junction pathways and regulated the testes growth and reproduction in bovines [69] and are also found in sperm morphogenesis to give rise to mature spermatozoon in mammals [70]. The transcript *SPO11* showed the highest expression in human fetal gonadal development [71]. Another study highlighted that the *INHBA* gene belongs to the major *TGFB* pathway and regulates early germ cell proliferation in rainbow trout testes [72]. Gene expressions are sex-specific in the sex development of mammals. Different genes express in different phases; correspondingly, the *FGF9* gene has been defined in male gonadal morphogenesis [73].

The past literature proposed that lncRNA expression can manipulate and highly interact with neighboring protein-coding gene expression via transcriptional coactivation [74,75]. Gao et al. [8] predicted the potential role of *cis*-target genes of differentially expressed lncRNAs in the regulation of testes development and spermatogenesis. In this study, we selected four differentially expressed lncRNA transcripts, along with their *cis*-target genes, for RT-qPCR validation. We found consistency in the expression results, and thus, the genetic regulatory network plays a vital role in cattle testes development and germ cell proliferation [76].

## 5. Conclusions

The lncRNAs and mRNAs were abundantly expressed in Wandong cattle testes development, and many of them were significantly differentially expressed between the calves and bulls. The lncRNAs may have significant biological functions in bovine testicular development. Thus, we thoroughly analyzed the GO descriptions and found that the source genes were enriched in gonad development, spermatid development, spermatogenesis, spermatid differentiation and the reproductive process. The lncRNA-linked genes *COX7A2, COX6B2, TRIM37, PRM2, INHBA, ERBB4, SDHA, ATP6VOA2, FGF9* and *TCF21* and mRNAs *CCDC83, DMRTC2, HSPA2, IQCG, PACRG, SPO11, EHHADH, SPP1, NSD2* and *ACTN4* were found to be actively associated with bull sexual maturity and spermatogenesis. The obtained results can provide a molecular fingerprint for assessing the developmental status of bull testes and also facilitate the selection of local elite bulls.

## Figures and Tables

**Figure 1 animals-11-02006-f001:**
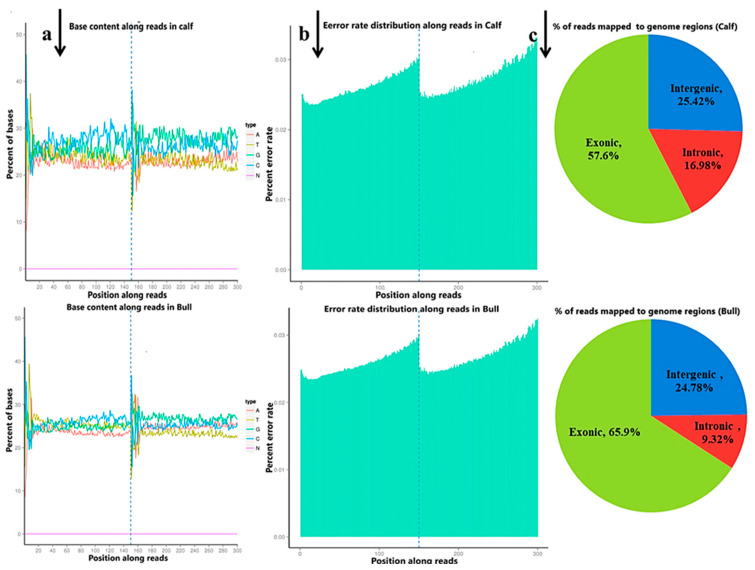
The quality summary of the clean reads used in the subsequent analysis were obtained after the original data filtering, sequencing error rate checking and GC content distribution checking. (**a**) The abscissa shows the base position of the reads, and the ordinate is the proportion of a single base, whereas different colors represent different base types. (**b**) The abscissa represents the bases position of the reads, and the ordinate is the average error rate of all the reads at this location. The 0~150 bp to the left of the dotted line is the error rate distribution of read1, while the 150~300 bp to the right is the error rate distribution of read2. (**c**) The percentage of the total number of clean reads mapped to the reference genome. The proportion of reads in the exon region, intron region and intergene region of the genome were calculated.

**Figure 2 animals-11-02006-f002:**
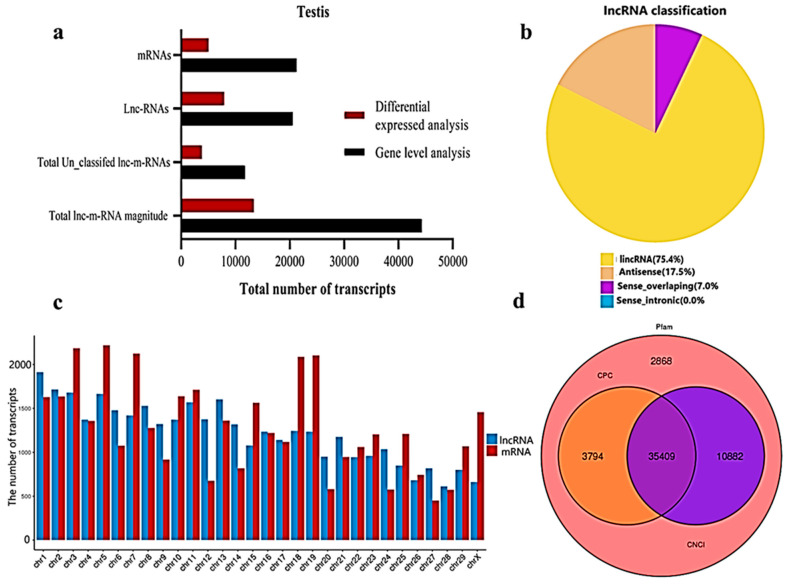
General properties of the lncRNAs and mRNAs in the premature and mature bovine testes were identified. (**a**) The transcripts were categorized as the total magnitude, unclassified and specified lncRNAs and mRNAs while analyzed at the gene level and differentially expressed patterns. (**b**) The distribution of lncRNAs at the genome region at a high magnitude were identified from the exon, intergenic and intron parts of the genome. (**c**) The distribution of testis mRNAs and lncRNAs in bovine chromosomal sets. (**d**) Venn diagram showing the coding potential results of the lncRNAs and intersection of the coding tools, such as CNCI, PFAM and CPC2.

**Figure 3 animals-11-02006-f003:**
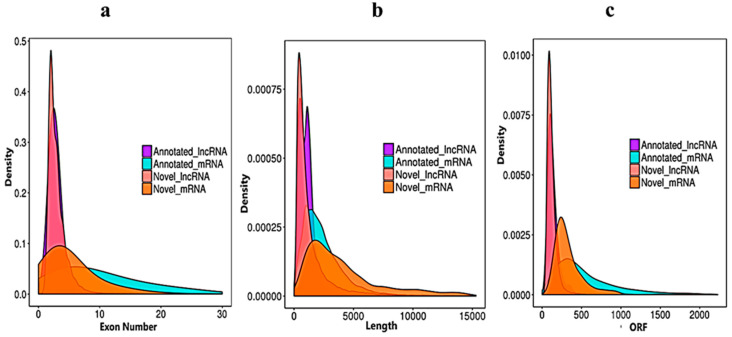
Sequencing characteristics of the lncRNAs and mRNAs have shown. (**a**) Exon number distribution of the annotated and novel lncRNAs (purple and pink) vs. the mRNAs denoted blue and dark yellow. (**b**) Transcript length comparisons of the annotated and novel lncRNAs (purple and pink) vs. the mRNAs (blue and dark yellow). (**c**) Open reading frame nucleotide sequence comparisons of the annotated and novel lncRNAs (purple and pink) vs. the mRNAs (blue and dark yellow).

**Figure 4 animals-11-02006-f004:**
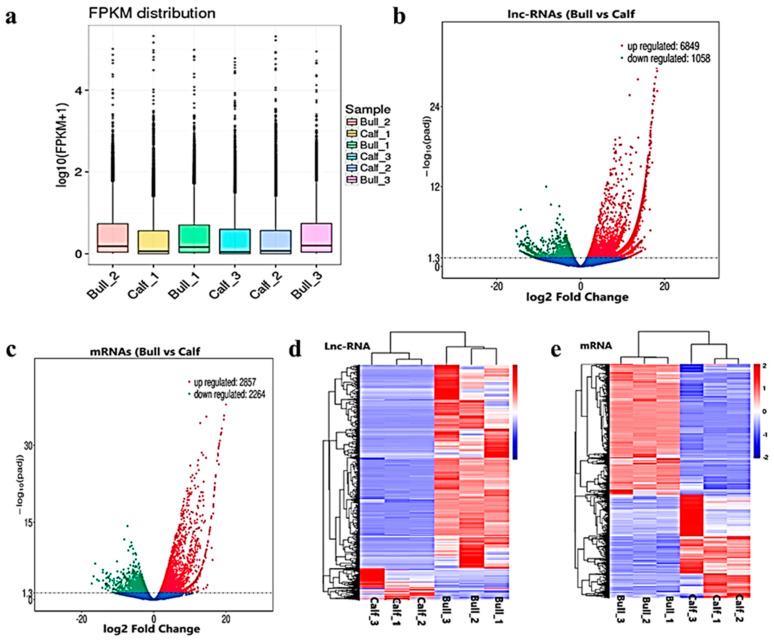
Differential expressed level of the transcripts and protein-coding genes in the testes. (**a**) The distribution of the expression levels of the genes or transcripts in different samples is shown by a box graph. The boxplot of each region corresponds to five statistics, e.g., the maximum, upper quartile, median, lower quartile and minimum, respectively. (**b**) The volcanic map visualizes the overall distribution of the lncRNAs genes with significant differences in their expression. (**c**) Differential mRNA expressions between the calves and bulls. The upregulated genes are represented by red dots, downregulated genes by green dots and the blue dots are genes without significant changes. (**d**) Clustering map of DE lncRNAs among the calf and bull testes sample. (**e**) Clustering map of the DE mRNAs among the samples. Red indicates upregulated coding genes, and blue indicates downregulated gene products.

**Figure 5 animals-11-02006-f005:**
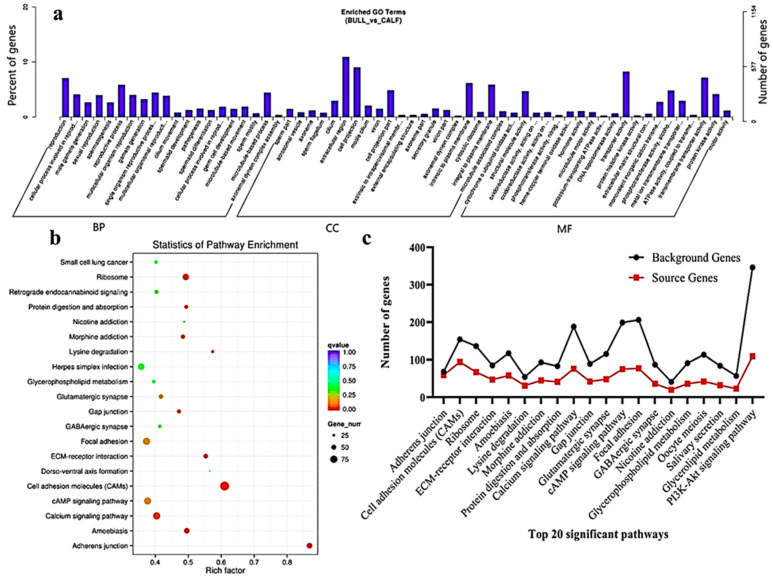
GO and KEGG functional validation of DE mRNAs. (**a**) GO enrichment analysis of DE mRNA genes between immature and mature Wandong cattle testes. DE mRNAs genes are divided into the following three biological segments: biological process (BP), cellular components (CC) and molecular function (MF), while the left and right *y*-axis showed the percentage and numbers of mRNA host genes. (**b**) Top 20 enriched pathways of DE mRNA host genes in the immature and mature phases of testicular growth. (**c**) Top 20 most relevant pathways for reproduction and representing the total background genes and significant source genes of mRNAs in the pathways.

**Figure 6 animals-11-02006-f006:**
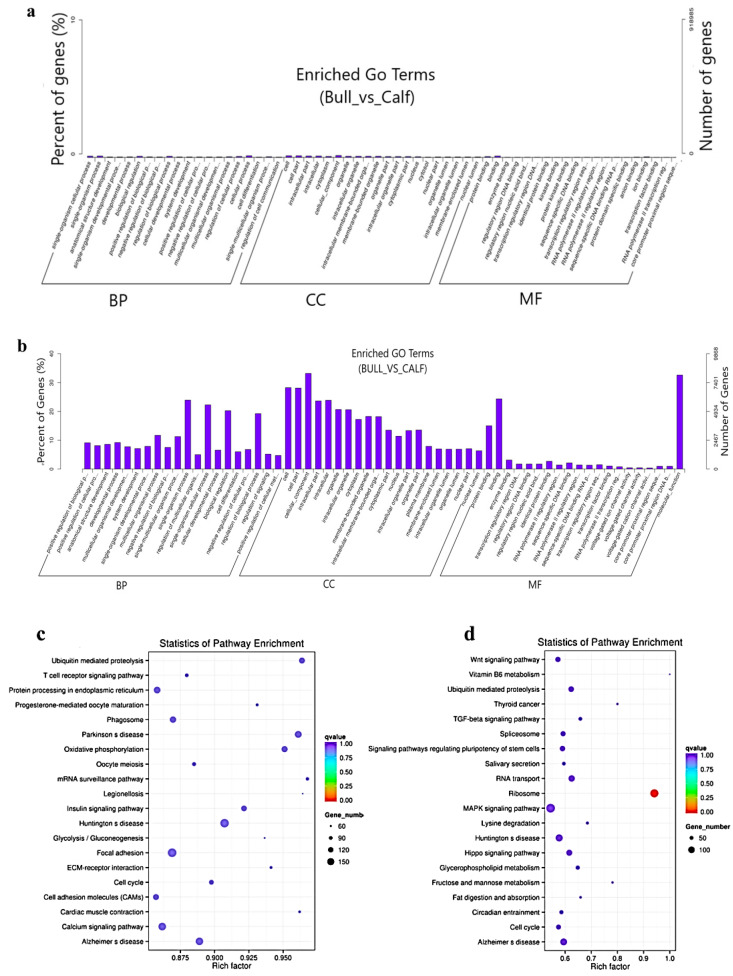
The co-location and co-expression methods were used and identified the lncRNA *cis*- and *trans*-regulated genes. (**a**) The scheme represents the co-expressed (*trans*-regulated) genes of the lncRNAs. (**b**) Co-located (*cis*-regulated) genes of the lncRNAs. (**c**) The top 20 significant enriched KEGG pathways for co-expressed lncRNA target genes are listed. (**d**) The top 20 significant enriched KEGG pathways for co-located lncRNA target genes are listed.

**Figure 7 animals-11-02006-f007:**
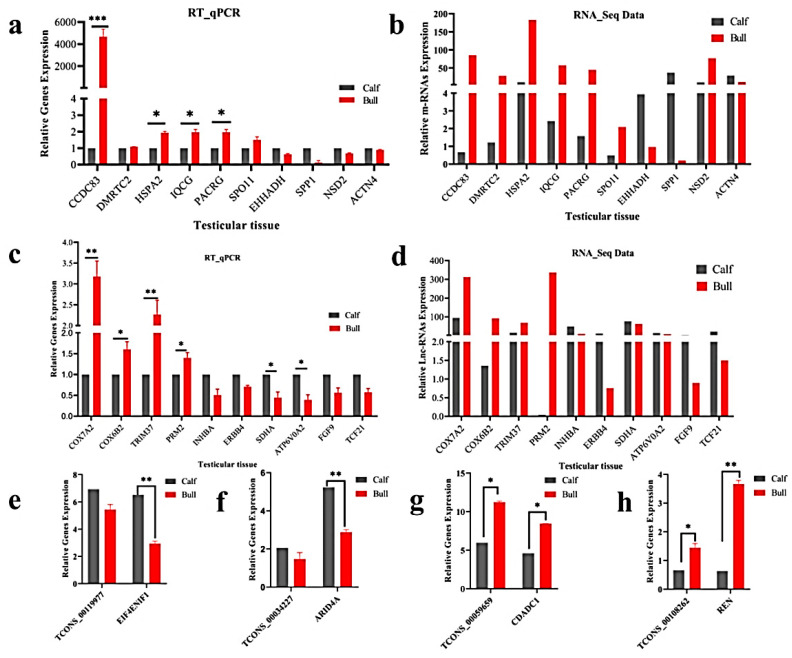
Validation of the RNA-Seq results via RT-qPCR in the immature and mature stages of bulls. The penal figures (**a**–**d**) compared the DE mRNA and lncRNA sequencing results and the RT-qPCR findings. The penal figures (**e**–**h**) showed the expression decoration of four lncRNAs and their *cis*-acting genes in two different developmental stages. The data were measured by the 2−ΔΔCt method, and *GAPDH* was used as the housekeeping gene. The data are presented as the mean ± SEM, where * *p* < 0.05, ** *p* < 0.01, *** *p* < 0.001.

**Table 1 animals-11-02006-t001:** Top 20 DE *cis*-acting long non-coding RNAs in calf and bull testes.

DE-lnc-RNA/ID	Chromosome/Location	Lnc-RNA-Status	Regulation	DE *cis*-Regulated Genes/ID	Gene-Symbol	Chromosome/Location	Enrich Pathways
TCONS_00223595	25: 40984697– 41226938	Novel-lncRNA	Up	ENSBTAG00000002474	*MAD1L1*	25: 40984696– 41222673	Cell Cycle
TCONS_00218376	24: 47452790– 47459159	Novel-lncRNA	Down	ENSBTAG00000039916	*SMAD2*	24: 47500431– 47586992	Cell Cycle
TCONS_00285708	4: 49484422– 49563305	Novel-lncRNA	Down	ENSBTAG00000006732	*NRCAM*	4: 49248702– 49563305	Cell Adhesion Molecules (CAMs)
TCONS_00265960	3: 3819912– 3824649	Novel-lncRNA	Down	ENSBTAG00000012025	*LMX1A*	3: 3653710– 3828022	Cell Adhesion Molecules (CAMs)
TCONS_00263842	3: 1.02 × 10^8^– 1.02 × 10^8^	Novel-lncRNA	Up	ENSBTAG00000000253	*PTPRF*	3: 102414018–102498351	Adherens Junction
TCONS_00092191	15: 30147243– 30156216	Novel-lncRNA	Down	ENSBTAG00000015939	*NECTIN1*	15: 30211169– 30282836	Adherens Junction
TCONS_00226910	25: 35113818– 35117275	Novel-lncRNA	Up	ENSBTAG00000026273	*MYL10*	25: 35177976– 35190739	Tight Junction
TCONS_00003412	1: 53213665– 53221154	Novel-lncRNA	Up	ENSBTAG00000018399	*MYH15*	1: 53114698– 53273181	Tight Junction
TCONS_00314718	6: 98071217– 98093465	Novel-lncRNA	Down	ENSBTAG00000005745	*HPSE*	6: 98071126– 98118656	Metabolic Pathway
TCONS_00305722	5: 76121498– 76122025	Novel-lncRNA	Down	ENSBTAG00000030632	*ALG10*	5: 76104065– 76113301	Metabolic Pathway
TCONS_00106232	16: 56589861– 56596165	Novel-lncRNA	Down	ENSBTAG00000011706	*TNR*	16: 56545152– 56764975	ECM-receptor Interaction
TCONS_00340198	7: 75125739– 75137127	Novel-lncRNA	Up	ENSBTAG00000014773	*HMMR*	7: 75137017– 75184925	ECM-receptor Interaction
TCONS_00282996	4: 11670810– 11680838	Novel-lncRNA	Down	ENSBTAG00000013472	*COL1A2*	4: 11776162– 11823181	Focal Adhesion
ENSBTAT00000080812	5: 32333604– 32349222	Annotated-lncRNA	Up	ENSBTAG00000013155	*COL2A1*	5: 32283180– 32313172	Focal Adhesion
TCONS_00098825	15: 37889979– 37909497	Novel-lncRNA	Up	ENSBTAG00000005218	*PDE3B*	15: 37935445– 38104280	cAMP Signaling Pathway
TCONS_00267075	3: 16683665– 16685238	Novel-lncRNA	Up	ENSBTAG00000006234	*NPR1*	3: 16683665– 16700685	cAMP Signaling Pathway
TCONS_00118360	17: 53611398– 53625902	Novel-lncRNA	Down	ENSBTAG00000004457	*ORAI1*	17: 53610136– 53626039	Calcium Signaling Pathway
TCONS_00038312	11: 22365812– 22406197	Novel-lncRNA	Down	ENSBTAG00000013861	*SLC8A1*	11: 22408368– 22787993	Calcium Signaling Pathway
TCONS_00310078	6: 11726041– 11732204	Novel-lncRNA	Down	ENSBTAG00000014463	*CAMK2D*	6: 11800357– 12189732	GnRH Signaling Pathway
TCONS_00321828	6: 1.01 × 10^8^– 1.01 × 10^8^	Novel-lncRNA	Down	ENSBTAG00000020048	*MAPK10*	6: 100907869–101279063	GnRH Signaling Pathway

## Data Availability

All data generated or analyzed during this study are available from the corresponding authors upon reasonable request.

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
