# Peer review of "Integrated Analysis of Long Non-Coding RNA and mRNA Expression Profiles in Testes of Calves and Sexually Mature Wandong Bulls (Bos taurus)"

_animals, 2021, doi:10.3390/ani11072006_

Round 1
Reviewer 1 Report
All comments were corrected.
Author Response
Riviewer1
1. All comments were corrected.
Response: Thank you very much for the good comments.
Reviewer 2 Report
First of all I want to thank the authors for their efforts in revising the manuscript that in the present version is now more understandable for the readers. In my opinion it is now acceptable for the publication.
Author Response
Reviewer 2
- First of all I want to thank the authors for their efforts in revising the manuscript that in the present version is now more understandable for the readers. In my opinion it is now acceptable for the publication.
- Response: Thank you very much for the nice comments.
Reviewer 3 Report
Excellent work! The exposure is clear and fluent, the methodological approach comprehensive and thorough, the results are very interesting.
Author Response
Reviewer 3
- Excellent work! The exposure is clear and fluent, the methodological approach comprehensive and thorough, the results are very interesting.
- Response: Thank you very much for the nice comments.
This manuscript is a resubmission of an earlier submission. The following is a list of the peer review reports and author responses from that submission.
Round 1
Reviewer 1 Report
The study fromshows an existence of interactive mRNAs and lncRNAs network which, at the molecular level, is involved in the regulation of spematogenesis in bull testesThe study (manuscript ID: animals-1214294) from Liu et al., shows an existence of interactive mRNAs and lncRNAs network which, at the molecular level, is involved in the regulation of spematogenesis in bull testes. Study is interesting and provides valuable data , it requiresan improvemen in several aspects.
1.Working hypothesis and aim of the study are missing in the manuscript. Research was performed on two selected and age-different groups of animals. In the Introduction paragraph, however, there is no clear explanation concerning the rationale of their experimental choice as well as whether transcriptomic profiles differences between those grousp were expected.
- The present version of the Statistical analyses paragraph, suggest (however it is not clearly stated) that also for qPCR results analysis a parametric t-test has been chosen. Such choise raises doubts due to the extremly low abundance of experimental groups (n=3). In fact nonparametric rank-test appears of special usefulness to analyze data on relative mRNA expression levels that were obtained by comparative quantitative method.
3.Discussion paragraph would be improved with more detailed focus concerning differences between two analyzed groups. For example statements like: ‘(…) we identified 6849 lncRNAs significantly upregulated and 2264 were down-regulated between two groups (…) does not inform about association with animals age (how many lnc RNAs and mRNAs were up/down- regulated in each group).
- A serious editorial shortcoming of the manuscript is due to its figures illegibility in Acrobat version. In fact, Authors should keep in mind that some potential readers may prefer printed version of the paper to be studied. Therefore, to fulfill a required editorial standard, a comprehensive graphic revision (with special focus on fonts size) of figures is indispensable.
- A careful point-to point examination concerning the descriptions given on the figures vs descriptions given in the legends is necessary since some details appear confusing For example on Fig 3 b a given description of the (predominantly) yellow graph is „lncRNA classification”, whereas the legend to Fig 3b there is informs that it refers to „ distribution of circRNAs”. Moreover, why letter „b” was introduced twicely into the legend concerning Figure 3?
- Why SEM values are not included on graphs concerning qPCR results obtained on calf group and and why only on some graphs from bulls group the specific SEM value is present (for both see Fig 8)? Whereas when referring to the Figure 8 legend, a reader can find an explanation that „the data are present as the MEAN ± SEM”
- It would be advisable to improve the English edition of the manuscript.
Author Response
Dear Reviewer!
Please find the attached file having our point by point response of your kind comments.

Reviewer 2 Report
The article describes an extensive Analysis of mRNA and long non-coding RNA Expression Profiling Differences Associated with Testicles Development in Chinese local cattle (Bos taurus). Despite the topic is interesting to those with closely related research interests, there are few issues that need further consideration.
#Minor
Page 14 – “…..suggested that the correlation coefficient should be greater than 0.95.”
Page 15 – In the sentence: “The findings of the GO enrichment study revealed that there were 70 GO description were significantly observed (P-adj-value < 0.05),…” remove the second “were”
Author Response
Dear Reviewer!
Please find the attached file, which containing our point by point response of your kind comments.

Reviewer 3 Report
The authors presented a paper on extensive analysis of mRNA and long non-coding RNA expression profiling differences associated with testicle development in Chinese cattle.
The work is well conducted. In my opinion the paper is worthy of publication even if several revisions are requested before to further process the manuscript.
General comment: please include again the number of lines as in the MDPI template file. The absence of line numbers makes the reviewer work so strong. For this reason all my following comments are referring onyl to the page number.
The material and methods section is quite strong to read. I suggest to considerably summarize this section that is really too long.
In the abstract section try to avoid, where possible, the use of acronyms.
More in details:
keywords: please try to avoid words already included in the title.
-page 2: it could be better “and spermatogenesis of local superior breeds remains….”
-page 2: this sentence needs a reference “The interactive network of long noncoding-RNA and mRNA thought to be involved in mammalian reproduction system regulation,and sperm quality parameters.”
-page 3: for the gene acronyms “genes Mrh1, Tsx, Dmr, and HongrES2” is better to use capital letters. Moreover, the first time could be useful to include the gene full name.
-page 3, material and methods: no information are supplied on the studied breed. Why the authors studied this breed? It could be important to adequately describe the importance of this breed for the local economy. Moreover, are there information about the sperm quality of the 3 mature bulls?
-page 3: it could be better “(3± 0.014 years)” and “(3± 0.24 months)”.
-page 3: it is better “at Fengyang County, Anhui, China”.
-page 3: even if well known please specify the temperature of the liquid nitrogen.
-page 3, section 2.2. Histological Study of Testicular Tissues, please specify the magnification level used for the histological study and the kind of microscope used.
-page 4: “templet strand” or “template strand”?
-page 4: please check the sentence “AM-Pure XP beads. Finally! The USER enzyme…” The word Finally! seems to be out the context.
-page 5: “than 200 base pairs (bp) were..” Try to use the acronym bp starting from here.
-page 5: “were also discorded” or “were also discarded”?
-page 6: “(P-value <0.05)”, please use the italic style (and capital letter) for P and be consistent in all the manuscript long.
-page 6: “KOBAS software”, please specify the version.
-page 7: “Whole RNA was extracted from….”: starting from how many mg of tissue? Please specify.
-page 7: “genes between 100k upstream and downstream”, please specify the reason to justify this interval. Any reference?
-page 8 “clean reads (70333218, 98.95%)”: please specify the first time the meaning of these values.
-page 8, please replace “to the B. taurus” with “to the Bos taurus”.
-figure 8: “UMD3.1”: any available reference?
-page 10, Figure 3d: please check the legend. Some brackets are missing.
-page 11, Figure 4: it is not easy to appreciate the differences because the 3 graphs have differente scale in the axes (especially Y axe). It could be usueful to use a different graphical representation.
-page 11: the sentence “The average transcripts expression level is higher in bulls comparing to calves group..” is not so evident in the Figure 5a. Please check. Moreover, the Figure 5 needs to be revised: -brackets are missing; - what is the legend meaning in figure 5d?
-page 15, Table 1, please use italic style for all the gene symbols.
-page 16: Figure 7: especially the images a and b are really too small to be correctly understood. In this way the images are really unuseful for the readers.
-page 17: “are available at (Supplementary files)”, please specify the number.
-page 19: “Generally! We” The word Generally! seems to be out the context.
-page 20: please pay attention to the gene acronyms, sometime in italic style sometime not.
The conclusions section is too much generalistic. Starting from your results try to include a clear message for the readers.
Supplementary file 5: in the title please replace “summery” with “summary”.
Author Response

(The authors gave the same response as above.)

Reviewer 4 Report
The work is very interesting and well described, in the attached file you will find some suggestions and minor revisions.

Author Response
Dear Reviewer!
Please find the attached file containing our point by point response of your kind comments.
